# NCAPG Promotes Pulmonary Artery Smooth Muscle Cell Proliferation as a Promising Therapeutic Target of Idiopathic Pulmonary Hypertension: Bioinformatics Analysis and Experiment Verification

**DOI:** 10.3390/ijms231911762

**Published:** 2022-10-04

**Authors:** Bowen Fu, You Li, Xiaobo Shi, Peng Liu, Yiman Zhang, Hongyan Tian

**Affiliations:** 1Department of Peripheral Vascular, The First Affiliated Hospital of Xi’an Jiaotong University, Xi’an 710061, China; 2Department of Radiation Oncology, The Second Affiliated Hospital of Xi’an Jiaotong University, Xi’an 710004, China; 3Department of Cardiovascular Medicine, The Second Affiliated Hospital of Xi’an Jiaotong University, Xi’an 710004, China

**Keywords:** pulmonary arterial hypertension, bioinformatics, differentially expressed genes, NCAPG, proliferation

## Abstract

Idiopathic pulmonary arterial hypertension (IPAH) is a disease with complex etiology. Currently, IPAH treatment is limited, and patients’ prognosis is poor. This study aimed to explore new therapeutic targets in IPAH through bioinformatics. Two data sets (GSE113439 and GSE130391) meeting the requirements were obtained from the Gene Expression Omnibus (GEO) database. Then, differentially expressed genes (DEGs) were identified and analyzed by NetworkAnalyst platform. By enriching Gene Ontology (GO) and the Kyoto Encyclopedia of Genes and Genomes (KEGG), we examined the function of DEGs. A protein–protein interaction (PPI) network was constructed to identify central genes using the CytoNCA plug-in. Finally, four central genes, *ASPM*, *CENPE*, *NCAPG*, and *TOP2A*, were screened out. We selected *NCAPG* for protein-level verification. We established an animal model of PAH and found that the expression of NCAPG was significantly increased in the lung tissue of PAH rats. In vitro experiments showed that the expression of NCAPG was significantly increased in proliferative pulmonary arterial smooth muscle cells (PASMCs). When *NCAPG* of PASMCs was knocked down, the cell proliferation was inhibited, which suggested that *NCAPG* was related to the proliferation of PASMCs. Therefore, these results may provide new therapeutic targets for IPAH.

## 1. Introduction

Pulmonary arterial hypertension (PAH) is a clinical syndrome with changes in pulmonary vascular structure or function caused by a variety of reasons, resulting in increased pulmonary vascular resistance and pulmonary artery pressure, eventually leading to right heart failure and even death. Worldwide, the incidence of PAH is as high as 7.6 cases per million adults and the prevalence ranges from 26 to 100 cases per million adults [1]. The European Society of Cardiology (ESC) guidelines classify PAH into five types [2], and idiopathic pulmonary arterial hypertension (IPAH) is one of the subtypes with unclear causes. IPAH is the most common subtype of PAH in Western countries, accounting for 30 to 50% of all PAH cases [3]. The pathological changes of IPAH involve multiple mechanisms, which are still unclear [4]. In the era of traditional therapy without targeted drugs, IPAH has a poor prognosis, with a median survival time of only 2.8 years [5]. With the application of targeted drugs, the 5-year survival rate of newly diagnosed PAH patients has increased to 61.2% [6]. However, a single-center study in China showed that IPAH, a common type of PAH in childhood, had 1-, 3-, and 5-year survival rates of 53.5%, 46.5%, and 31.2%, respectively [7]. At present, the main drugs used in the treatment of IPAH are prostacyclin analogues and receptor agonists, phosphodiesterase type 5 inhibitors, endothelin receptor antagonists, and targeted drugs such as riociguat, whose main effect is to dilate pulmonary blood vessels. Drugs targeting other pathologic mechanisms, such as pulmonary artery remodeling, are lacking [8]. In addition, the long-term use of these drugs is expensive, and low-income people cannot afford them. As a chronic malignant disease, IPAH will bring a heavy burden to society and the family if it is not well treated. Therefore, it is urgent to explore novel molecular mechanisms and potential therapeutic targets for IPAH.

In the field of oncology, the use of microarray and high-throughput sequencing technologies to predict potential targets for diseases is well established [9,10]. In recent years, this method has also been used to detect PAH patients, animals, or cell samples and analyze and screen the corresponding gene therapy targets [11,12]. The Gene Expression Omnibus (GEO) database provides gene expression microarray data of multiple groups of IPAH tissues or cells. The use of bioinformatics analysis methods can help us screen potential genes related to the pathogenesis of IPAH and provide certain guidance for targeted therapy. Bai Z et al. identified two central genes, ECM2 and GLT8D2, associated with PAH by using the weighted gene coexpression network analysis (WGCNA) algorithm [13]. Lin W et al. found that TXNRD1 as a novel biomarker and potential therapeutic target for IPAH through Robust Rank Aggregation (RRA) method and experiments [14]. However, some of the current studies have the shortcomings of only making theoretical predictions and not further verifying the reliability of their prediction results through experiments [15,16].

The sample size of a single microarray is small, and there is a problem that the false positive rate of the analysis results is high. In this study, GSE113439 and GSE130391, two independent microarray data sets containing IPAH patient samples, were selected from GEO database for analysis. We identified 612 differentially expressed genes (DEGs) and analyzed their functions through Gene Ontology (GO) annotation and Kyoto Encyclopedia of Genes and Genomes (KEGG) pathway enrichment. Four central genes were screened by establishing protein–protein interaction (PPI) network, and their mRNA expression was verified by quantitative reverse transcription-polymerase chain reaction (qRT-PCR). Through further in vivo and in vitro experiments, we finally identified *NCAPG* as a potential therapeutic target for IPAH.

## 2. Results

### 2.1. Identification of DEGs

Two gene expression data sets, GSE113439 and GSE130391, were selected for this study. We selected, normalized, and annotated data from IPAH patients and healthy controls. The preprocessed data set was uploaded to the NetworkAnalyst platform to identify DEGs. There was a total of 3834 DEGs in the GSE113439 data set, including 2449 upregulated genes and 1385 downregulated genes (Appendix A). There was a total of 3524 DEGs in the GSE130391 data set, including 1336 upregulated genes and 2188 downregulated genes (Appendix A). The intersection DEGs of the two data sets were screened by the Venny 2.1 platform, and there were 394 upregulated DEGs (Appendix A) and 218 downregulated DEGs (Appendix A). Figure 1 shows the volcano and Venn plots for the DEGs in each data set. A heatmap of 612 DEGs is shown in Appendix A.

### 2.2. Enrichment Analysis of DEGs

We performed GO enrichment analysis and KEGG pathway enrichment analysis on the previously selected DEGs to evaluate the biological roles of these genes in IPAH. The top 10 enriched GO terms for biological process (BP), cellular component (CC), and molecular function (MF) for 612 DEGs are shown in Figure 2A. BP analysis was mainly enriched in mRNA processing, RNA splicing, and regulation of GTPase activity, among others. CC analysis was mainly enriched in the chromosomal region, nuclear speck, and spindle, among others. MF analysis was mainly enriched in catalytic activity, acting on DNA, histone binding, and helicase activity, among others. In the KEGG pathway analysis (Figure 2B), genes mainly involved in spliceosome, endocytosis, and human papillomavirus infection were enriched. Appendix A shows the detailed results of the enrichment analysis.

### 2.3. PPI Network Establishment and Central Genes Identification

To construct the PPI network, 612 DEGs were incorporated. The PPI network consisted of 213 nodes and 366 edges with the highest confidence score (0.900) based on the STRING database. The remaining 399 DEGs were not suitable for the final PPI network. We used the CytoNCA plug-in to calculate topology parameter information based on the topology properties of network nodes, including betweenness centrality (BC), closeness centrality (CC), degree (DC), eigenvector centrality (EC), local average connectivity (LAC), and network centrality (NC) (Appendix A). Finally, the four target proteins ASPM, CENPE, NCAPG, and TOP2A were determined to be the central nodes of the entire network (Figure 3).

### 2.4. Validation of Selected Candidate Genes in Animal Models of PAH

Because it is difficult to obtain lung tissue samples from patients with IPAH in clinical practice, we established the monocrotaline (MCT)-induced PAH rat model for further experiments, which is a recognized animal model of PAH. After measurement, the right ventricular systolic blood pressure (RVSP) and RV/(LV + S) of MCT-induced PAH rats were significantly higher than those of the control group (Figure 4A). Hematoxylin and eosin (HE) staining also showed significant pulmonary artery remodeling (Figure 4B). We performed qRT-PCR analysis using rat lung tissue to validate the screened central genes. The results of qRT-PCR showed that compared with the control group, there were significant differences in the expression of four genes (*ASPM*, *CENPE*, *NCAPG*, and *TOP2A*) in the PAH group (Figure 4C).

### 2.5. NACPG Is Correlated with Pulmonary Artery Smooth Muscle Cell Proliferation

We selected *NCAPG* with relatively high expression among the four central genes for further verification. The lung tissue homogenate of rats was used for the Western blot experiment, and the results showed that the expression of NCAPG in the lung tissue of the PAH group was significantly increased compared with the control group (Figure 5A). Pulmonary artery smooth muscle cell (PASMCs) proliferation is one of the important mechanisms of pulmonary artery remodeling in PAH. Primary PASMCs were extracted from rats and identified by α-SMA immunofluorescence (Appendix A). Platelet-derived growth factor-BB (PDGF-BB) is a classical cytokine of pulmonary artery remodeling in PAH. We used PDGF-BB on PASMCs to induce their proliferation, and it was observed that the expression of PCNA, a marker of proliferation, was increased, and the expression of NCAPG was also increased, which was consistent with animal models (Figure 5B). Cell Counting Kit 8 (CCK8) results indicated that PASMCs proliferation was successfully induced (Figure 5C). NCAPG-targeting small interfering RNA (siRNA) was used to target silence *NCAPG* in PASMCs to further confirm the role of *NCAPG* in PASMCs (Figure 5D). The results showed that when the expression of NCAPG was downregulated, the expression of proliferation marker protein PCNA was also downregulated (Figure 5E). CCK8 assay showed that *NCAPG* silencing inhibited PASMC proliferation (Figure 5F). In conclusion, these results suggested that *NACPG* was correlated with PASMCs proliferation.

## 3. Discussion

PAH is a chronic disease with complex etiology. At present, the main treatment methods are drug therapy, surgery, and lung transplantation. With the application of targeted drugs, such as riociguat, the 5-year survival rate of newly diagnosed PAH patients has been improved to 61.2% [6]. However, there are still many patients without effective treatment, causing substantial economic and social burdens. At present, the pathogenesis of PAH is still not completely clear. With the development of genomic technology, some possible causes have been revealed. Some PAH gene mutations, such as the gene encoding bone morphogenetic protein type 2 receptor (BMPR2) protein, have been identified by whole-genome sequencing [17]. A recent study showed that unsupervised machine learning analysis of the whole blood transcriptome of IPAH patients could divide 92% of IPAH patients into three subgroups, each with unique whole blood transcriptomic and clinical features associated with prognosis [18]. These studies may indicate genetic differences in PAH. In recent years, the high-throughput omics method has made great achievements in the field of oncology research. More and more researchers have extended this method to the field of PAH and achieved some important results [11,12]. However, most of these studies are limited to the theoretical stage without experimental verification [15,16].

In this study, we selected two data sets containing IPAH samples from the GEO database for bioinformatics analysis (GSE113439 and GSE130391). These two data sets have not been previously analyzed jointly. By intersection, 612 DEGs were identified, of which 394 were upregulated, and 218 were downregulated. These DEGs were significantly enriched in processes related to DNA replication and transcription according to GO and KEGG enrichment analysis. After constructing the PPI network, we used the CytoNCA plug-in to screen for central genes. CytoNCA is an entirely new centrality tool to offer such comprehensive calculations, analyses, and evaluations for biological networks and has been cited in many studies [19,20]. It supports up to eight different centrality measures, making it more comprehensive than plug-ins such as CentiLib and NetworkAnalyzer. By including topological parameters BC, CC, DC, EC, LAC, and NC for multiple rounds of analysis, four central genes were screened out and then verified by qRT-PCR. Finally, *NCAPG* was selected for further verification. We established an animal model of PAH and found that the expression of NCAPG was significantly increased in the lung tissue of PAH rats, which was consistent with the predicted results. In vitro experiments showed that the expression of NCAPG was significantly increased in proliferative PASMCs. When *NCAPG* of PASMCs was knocked down, the cell proliferation was inhibited, which indicated that *NCAPG* was related to the proliferation of PASMCs. This study was the first to demonstrate *NCAPG* as a potential therapeutic target for IPAH.

Four important hub genes, *ASPM*, *CENPE*, *NCAPG*, and *TOP2A*, were identified through bioinformatics analysis and screening, all of which were upregulated. Centromere protein E (CENPE) plays an important role in cell mitosis, where it is enriched in the G2 phase of the cell cycle and consumed during mitosis [21]. Researchers have found that CENPE is involved in the regulating cell proliferation, apoptosis, and migration in tumor cells [22]. Studies have demonstrated that CENPE can promote pulmonary artery remodeling in PAH [23]. The lack of CENPE increased the mitochondrial apoptotic proteins Bim and cleaved caspase 3/9, thereby reversing hypoxia-induced PASMCs apoptosis resistance. Additionally, CENPE regulates the migration of PASMCs by regulating the expression of MMP4/9. Our qRT-PCR results indicated that the transcript level of *CENPE* in the lung tissue of PAH rats was significantly increased compared with that of the control group. This is consistent with previous findings [23].

Topoisomerase II alpha (TOP2A) mainly plays a role in DNA replication and mitosis [24]. When the cell cycle enters the S phase, TOP2A is involved in organizing genome structure, promoting chromosome segregation, and preventing aberrant entry into anaphase with partially decatenated sister chromatids [25]. Studies have shown that *TOP2A* is overexpressed in pancreatic cancer, breast cancer, and malignant peripheral schwannoma [26,27,28]. TOP2A can promote the malignant progression of lung adenocarcinoma cells and predict the poor prognosis of lung adenocarcinoma [29]. In the field of pulmonary hypertension, there is still a lack of studies on *TOP2A*. Our qRT-PCR results indicate that PAH rats have higher *TOP2A* expression in their lungs than healthy rats, which is consistent with previous studies [30]. Assembly factor for spindle microtubules (ASPM) is associated with the growth of various tumors, such as glioma, hepatocellular carcinoma, ovarian cancer, and pancreatic cancer [31,32,33,34]. Its mechanism might be related to regulating mitotic spindle function and promoting cell proliferation [35]. At present, the relationship between *ASPM* and PAH is still unclear. This study found that the expression of *ASPM* in the lung tissue of rats in the PAH group was increased, and we speculated that *ASPM* might be involved in the formation of PAH. PASMCs proliferation is one of the important pathological mechanisms of PAH, and both *TOP2A* and *ASPM* are related to cell mitosis. More studies are needed to explore the relationship between them and pulmonary artery remodeling in PAH. 

Non-SMC condensin I complex subunit G (NCAPG), a mitotic-associated chromosomal condensing protein, plays a crucial role in condensin activation and chromosome stability [36]. Recent studies have reported abnormal expression of *NCAPG* in gastric cancer, lung adenocarcinoma, prostate cancer, breast cancer, and hepatocellular carcinoma [37,38,39,40,41]. The relationship between *NCAPG* and PAH has not been studied before, and this is the first study to report it. We found that NCAPG was highly expressed in the lung tissue of PAH rats, as well as associated with the proliferation of PASMCs. Gong, C et al. found that *NCAPG* promoted the proliferation of hepatocellular carcinoma through the PI3K/AKT signaling pathway [41], and Wu, Y et al. found that *NCAPG* promoted the progression of lung adenocarcinoma through the transforming growth factor (TGF)-β signaling pathway [38]. The activation of the above two signaling pathways is also one of the mechanisms of PASMCs proliferation. We hypothesized that *NCAPG* might be involved in promoting PASMC proliferation, but further experiments are needed to confirm this. We found that downregulation of *NCAPG* inhibited PASMCs proliferation. Wu, Y. et al. screened out drugs targeting *NCAPG* such as fluorouracil, azathioprine, doxorubicin, and zoledronic acid [38]. Whether these drugs can be used in IPAH needs further experimental verification. This provides a new direction for the treatment of IPAH.

There are some limitations to our study. First, in this study, the selected data set species is *Homo sapiens*, and the sample is lung tissue. Lung tissue from patients with IPAH is the best choice for experimental validation. However, lung tissue samples of IPAH patients are usually from donation or organ transplantation and the sample size is rare. As a control, lung tissue from healthy people is more difficult to obtain and involves many ethical issues. It is really difficult to obtain human lung tissue samples to support our experiments in our current research conditions. Therefore, we chose to establish an animal model of PAH for verification. In order to maintain the consistency of experimental species before and after, we selected rat primary PASMCs for in vitro experiments. If sufficient human specimens and cells can be obtained in the future, we will continue to verify and improve the results. In addition, due to laboratory limitations, we only used the MCT-induced PAH model and PDGF-BB-induced PASMCs proliferation model, and if models were added, such as chronic hypoxia models, the results would be more convincing. Finally, we only found a correlation between *NCAPG* and PASMCs proliferation. The detailed mechanism of *NCAPG* promoting PASMCs proliferation deserves further investigation.

## 4. Materials and Methods

### 4.1. Data Collection and DEGs Identification

GEO is a public functional genomic data repository, and we used “pulmonary hypertension” and “microarray” as keywords to search for newly submitted transcriptomic datasets in the last 5 years, screening species as *Homo sapiens*, sample as lung tissue, target disease as IPAH. Finally, two data sets GSE113439 and GSE130391 were selected for further analysis. The information for the data sets is shown in Table 1. Their gene expression data sets and annotation files were obtained from the official website of GEO (https://www.ncbi.nlm.nih.gov/geo/, accessed on 20 August 2021). The microarray probes were annotated with annotation files and those mismatched gene probes were removed. If more than one probe corresponded to the same gene symbol, the average value was taken. The GSE113439 data set contained 6 patients with IPAH and 11 controls (the remaining 9 patients with other types of PAH were not used) [42]. The GSE130391 data set contained 4 patients with IPAH and 4 normal controls (the remaining 14 patients with chronic thromboembolic pulmonary hypertension were not used) [43]. DEGs analysis was performed using NetworkAnalyst (https://www.networkanalyst.ca/, accessed on 1 September 2021), an online analysis platform for comprehensive gene expression profiling and network visual analytics [44]. The preprocessed gene expression data were uploaded to the website, and the “limma” method was selected for analysis after data quality check. Genes with |log2 fold change (FC)| of ≥ 0.5 and adjusted *p*-value of <0.05 as basic parameters were selected as DEGs from each data set. *p*-Values were adjusted using false discovery rate (FDR). The target intersection set of the two gene data sets was obtained through the Venny 2.1 platform (https://bioinfogp.cnb.csic.es/tools/venny/, accessed on 2 September 2021).

### 4.2. GO Enrichment Analysis and KEGG Pathway Analysis

The GO enrichment analysis and KEGG pathway analysis were carried out using the R package “ClusterProfiler” [45]. Terms with an adjusted *p*-value of <0.05 were considered significant enrichment.

### 4.3. PPI Network Analysis

The previously selected DEGs were uploaded to the STRING online database (https://string-db.org/, accessed on 8 September 2021), which can provide analysis of interacting genes including physical and functional associations [46], to analyze and construct a PPI network. The PPI network was developed with the highest confidence score (0.900), and the generated networks were visualized and analyzed by Cytoscape 3.8.2 software (https://www.cytoscape.org, accessed on 8 September 2021). By using CytoNCA network topology analysis plug-in, PPI network topology structure was further analyzed with the help of topological parameters, such as BC, CC, DC, EC, LAC, and NC [47]. As all six topology parameters for the nodes were higher than the median value, they were considered to be crucial target proteins in the protein interaction networks.

### 4.4. PAH Animal Model

A total of 12 male Sprague–Dawley rats (200–220 g) (all purchased from the Laboratory Animal Center of Xi’an Jiaotong University) were used in this study. All experimental animal protocols used in this study were approved by the Biomedical Ethics Committee of Xi’an Jiaotong University Health Science Center. Rats were randomly divided into two groups: the control group (CON, *n* = 6) and the PAH group (PAH, *n* = 6). All rats were housed in the Laboratory Animal Center of Xi’an Jiaotong University with 12 h light/dark cycle (25 °C room temperature). Rodent chow and water were freely available. The experiment started after the rats were adapted to the environment. On the first day, MCT (60 mg/kg; Sigma-Aldrich, St. Louis, MO, USA) was intraperitoneally injected into six rats in the PAH group, and six rats in the CON group were intraperitoneally injected with physiological brine. After 28 days, the rats were anesthetized, and their RVSP was measured via jugular cannula. The heart and lung tissues were collected after the rats were sacrificed. RV hypertrophy index was calculated as $\mathrm{RV}\;\mathrm{hypertrophy}\;\mathrm{index}\;=\;\mathrm{RV}/\left(\mathrm{LV}\;+\;S\right)$, where RV is the right ventricle, LV is the left ventricle, and S is the septum.

### 4.5. Cell Culture Experiments

According to a previously reported method, primary PASMCs were isolated from rats [48]. Anesthetized rats were sacrificed, and lung tissue was rapidly isolated. The distal pulmonary artery was dissected, and connective tissue, adventitia, and intima were removed. To attach the pulmonary artery to the T25 culture flask wall, the artery was cut into pieces. The cells were cultured with Dulbecco’s Modified Eagle Medium (DMEM; Hyclone, Logan, UT, USA) containing 10% fetal bovine serum (FBS; Gemini, West Sacramento, CA, USA), 100 U/mL penicillin, and 100 μg/mL streptomycin (Hyclone, Logan, UT, USA) in a humidified atmosphere of 5% CO_2_ and 95% air at 37 °C. When the cells reached 80% confluence, they were passaged using 0.25% trypsin (Gibco, Grand Island, NY, USA). Cells at passages 3–5 were used throughout the experiments. PDGF-BB was used to induce PASMCs proliferation. Cells were pretreated with medium containing 1% FBS for 24 h and then treated with PDGF-BB (20 ng/mL; Peprotech, Rocky Hill, NJ, USA) for 24, 48, and 72 h. siRNA was synthesized by Hanheng Biotechnology (Shanghai, China). The following target sequences were used: si-NCAPG: 5′-GAUACAGAUUGUCACAGAA-3′. Based on the manufacturer’s instructions, the cells were transfected with si-NCAPG or non-target control siRNA in Opti-MEM medium (Gibco, Grand Island, NY, USA). The transfection reagent was Lipofectamine 2000 (Invitrogen, Carlsbad, CA, USA).

### 4.6. Western Blot Assay

Lung tissue or PASMCs was lysed using a radioimmunoprecipitation assay (RIPA) containing protease and phosphatase inhibitors (HEART, Xi’an, Shaanxi, China). After that, the lysate was centrifuged at 12,000 rpm for 15 min, and the supernatant was collected. To determine protein concentrations, the BCA Protein Assay Kit (Thermo Scientific, Rockford, IL, USA) was used. Protein samples were boiled with 5× loading buffer for 5 min and subjected to 10% sodium dodecyl sulfate-polyacrylamide gel electrophoresis (SDS-PAGE) and then transferred onto polyvinylidene difluoride (PVDF) membranes (Millipore, Billerica, MA, USA). Membranes were blocked with NcmBlot blocking buffer (NCM, Suzhou, Jiangsu, China) and incubated with primary antibodies at 4 °C overnight. Primary antibodies used in this study were: NCAPG (1:500, sc-515297, Santa Cruz Biotech, Dallas, TX, USA), PCNA (1:1000, #13110, Cell Signaling Technology, Danvers, MA, USA), and β-actin (1:2000, 20536-1-AP, Proteintech, Wuhan, Hubei, China). After washing the primary antibody with Tris-Buffered Saline and Tween 20 (TBST), incubation with horseradish peroxidase (HRP)-conjugated goat anti-rabbit (1:2500, bs-0295G-HRP, Bioss, Beijing, China) or goat anti-mouse immunoglobulin (Ig)G (1:2500, bs-0296G-HRP, Bioss, Beijing, China) for 1 h at room temperature was performed. The bands were detected using enhanced chemiluminescence with the ChemiDoc XRS System (Bio-Rad, Hercules, CA, USA) and analyzed with ImageJ 1.53m software (NIH, Bethesda, MD, USA).

### 4.7. RNA Extraction and qRT-PCR

Total RNA was extracted from rat lung tissue or PASMCs using a MiniBEST Universal RNA Extraction Kit (TaKaRa, Kusatsu, Shiga, Japan) according to the instructions. The concentration and quality of RNA were determined by the Eppendorf BioPhotometer D30 instrument (Eppendorf, Hamburg, Germany). After that, a PrimeScript™ RT Master Mix kit (TaKaRa, Kusatsu, Shiga, Japan) was used for the reverse transcription reaction. Subsequently, qRT-PCR was performed on the Bio-Rad CFX96 PCR System (Bio-Rad, Hercules, CA, USA) using FastStart™ Universal SYBR^®^ Green Master (Rox) (Roche, Mannheim, Germany). The primer sequences for four candidate genes (*ASPM*, *CENPE*, *NCAPG*, and *TOP2A*) are listed in Appendix A. Data were analyzed by the 2-DeltaDeltaCt method, and GAPDH was taken as an internal control.

### 4.8. CCK8 Assay

CCK8 (UElandy, Suzhou, Jiangsu, China) was used to evaluate cell proliferation. After si-NCAPG transfection or PDGF-BB intervention, the CCK8 reagent was added to the medium according to the manufacturer’s instructions. The absorbance at 450 nm was measured after 0, 24, 48, and 72 h.

### 4.9. Statistical Analysis

Statistical analysis was conducted using GraphPad Prism 9.0 software (GraphPad, San Diego, CA, USA). Differences between the two groups were analyzed with Student’s *t* test. Differences among multiple groups were evaluated using one-way ANOVA. Data were presented as mean ± standard error of the mean (SEM), and a *p* of <0.05 was considered statistically significant.

## 5. Conclusions

Four central genes (*ASPM*, *CENPE*, *NCAPG*, and *TOP2A*) associated with IPAH were identified by bioinformatics analysis in the GSE113439 and GSE130391 datasets. *NCAPG* was finally identified as a potential therapeutic target for IPAH through in vivo and in vitro experiments, which provided a new therapeutic idea for clinical work.

## Figures and Tables

**Figure 1 ijms-23-11762-f001:**
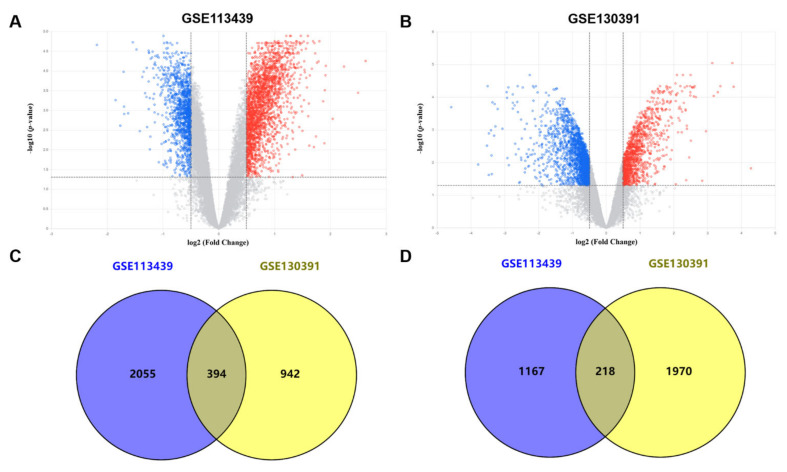
DEGs in microarray data sets GSE113439 and GSE130391. (**A**,**B**) Volcano plots of DEGs distributions for the two data sets. A red spot represents an upregulated DEG, and a blue spot represents a downregulated DEG, while a gray spot represents a non-DEG. (**C**) DEGs upregulated simultaneously in both data sets. (**D**) DEGs downregulated simultaneously in both data sets. DEG: differentially expressed gene.

**Figure 2 ijms-23-11762-f002:**
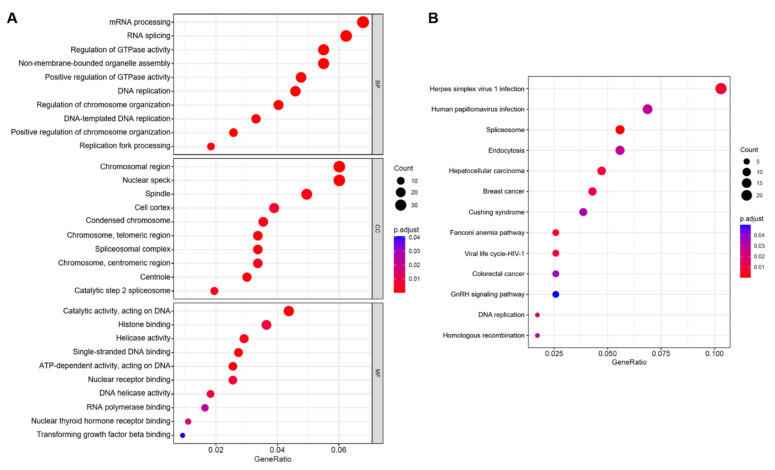
GO and KEGG enrichment analysis results of 612 overlapping DEGs. (**A**) The enriched GO terms consisted of biological process, cellular component, and molecular function. (**B**) KEGG analysis for DEGs. GO: Gene Ontology; KEGG: Kyoto Encyclopedia of Genes and Genomes; DEG: differentially expressed gene.

**Figure 3 ijms-23-11762-f003:**
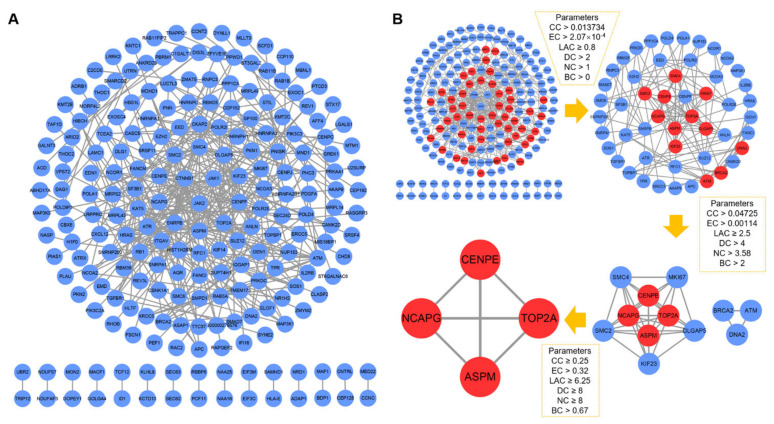
Establishment of the PPI network and identification of the central genes. (**A**) PPI network generated with the highest confidence score (0.900) on the STRING database. (**B**) CytoNCA plugin screened for central node genes. The red nodes represent the crucial targets of the entire network. PPI: protein–protein interaction.

**Figure 4 ijms-23-11762-f004:**
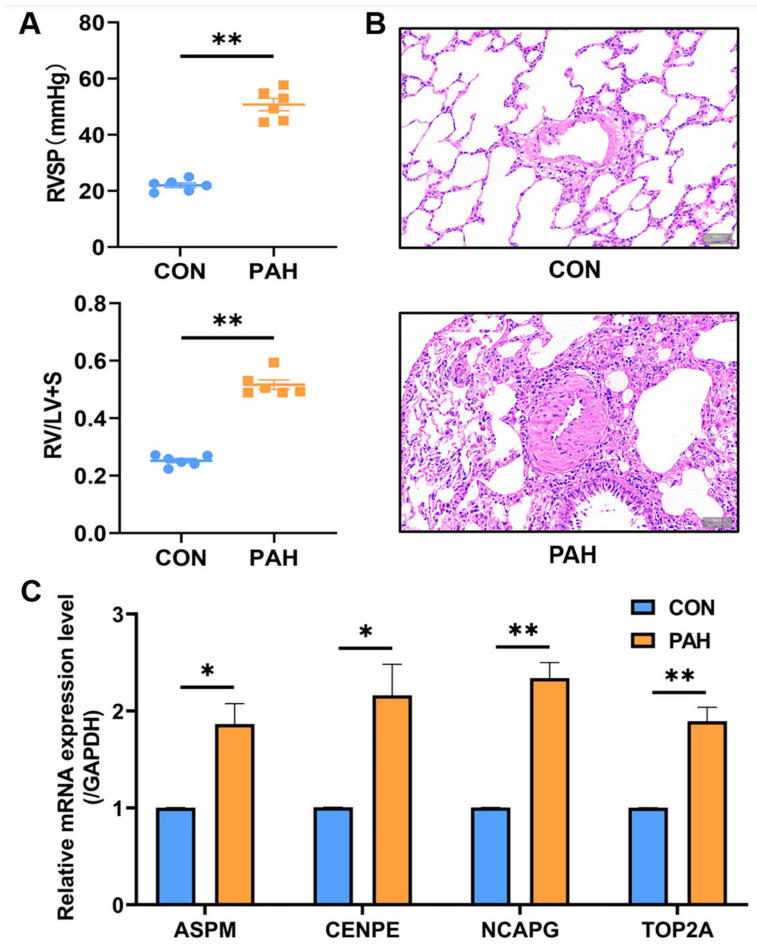
Validation of the central gene in animal models. (**A**) Hemodynamic data of lungs in animal experiments (*n* = 6 each). (**B**) Rat lung tissues stained with HE. (**C**) qRT-PCR was used to verify the expression of 4 central genes (*ASPM*, *CENPE*, *NCAPG*, *TOP2A*) (*n* = 6 each). * *p* < 0.05, ** *p* < 0.01.

**Figure 5 ijms-23-11762-f005:**
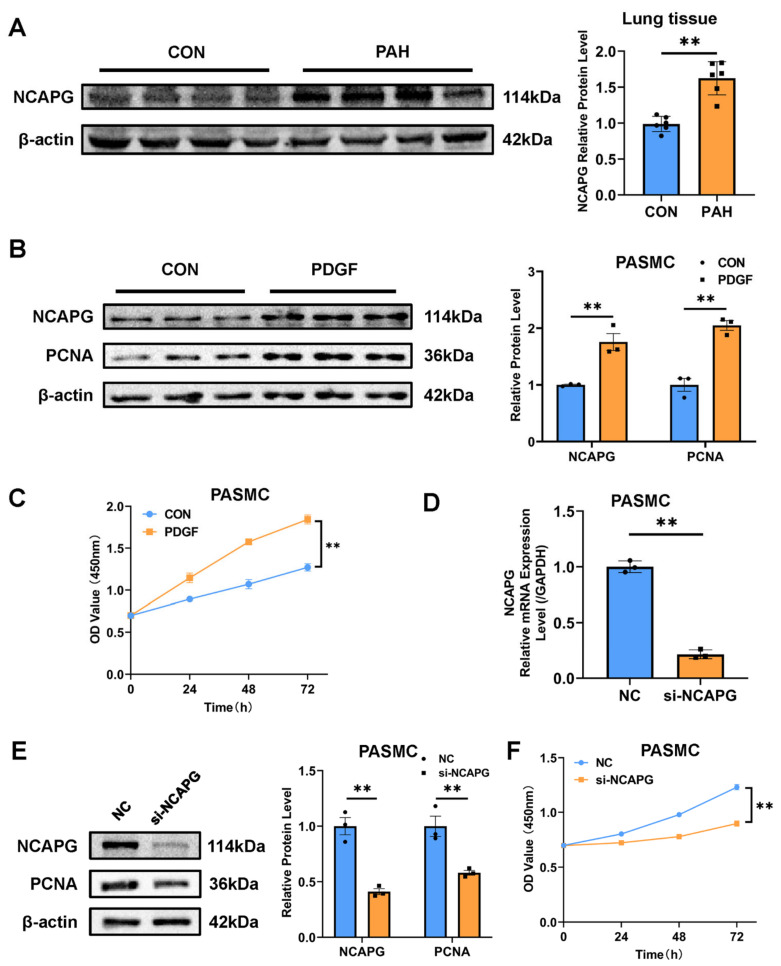
The expression of NCAPG was verified in vitro and in vivo. (**A**) The expression of NCAPG in rat lung tissues was analyzed by Western blot (*n* = 6 each). (**B**) The expression of NCAPG and PCNA in PASMCs stimulated by PDGF-BB (20 ng/mL) were analyzed by Western blot. (**C**) CCK8 assay was performed to determine the proliferative ability of PASMCs stimulated by PDGF-BB (20 ng/mL). (**D**) qRT-PCR was used to verify the knockdown efficiency of *NCAPG*. (**E**) Western blot was used to detect the protein expression level of PASMCs knocked down by *NCAPG*. (**F**) CCK8 assay was performed to determine the proliferative ability of PASMCs knocked down by *NCAPG*. PASMCs: pulmonary artery smooth muscle cells; PDGF-BB: platelet-derived growth factor-BB; CCK8: Cell Counting Kit-8. ** *p* < 0.01.

**Table 1 ijms-23-11762-t001:** Datasets detailed information.

References	Sample	GEO	Platform	IPAH	Control	Other (Not Using)
Mura, M. et al.	Lung tissue	GSE113439	GPL6244	6	11	9
Halliday, S. et al.	Lung tissue	GSE130391	GPL570	4	4	14

## Data Availability

Publicly available data sets were analyzed in this study. This data can be found here: GEO database (https://www.ncbi.nlm.nih.gov/geo/, accessed on 20 August 2021); the original contributions presented in the study are included in the article/Appendix A, and further inquiries can be directed to the corresponding author.

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
