# Peer review of "NCAPG Promotes Pulmonary Artery Smooth Muscle Cell Proliferation as a Promising Therapeutic Target of Idiopathic Pulmonary Hypertension: Bioinformatics Analysis and Experiment Verification"

_ijms, 2022, doi:10.3390/ijms231911762_

Round 1

Reviewer 2 Report

The manuscript of Fu et al. describes the examination of published microarray datasets in patients with idiopathic pulmonary arterial hypertension (IPAH) to identify novel genes involved in the pathogenesis. The authors used a number of bioinformatic approaches to narrow down the candidate genes, and arrived at 4 possible candidates. The authors were able to show a correlation between the expression of NCAPG mRNA and protein levels with IPAH. Using a cell model (pulmonary artery smooth muscle cells), the authors show that knocking down NCAPG levels decreased proliferation rates. While the evidence provided does suggest that NCAPG could be a gene of interest, the authors could have done a lot more to prove its involvement. I have a number of concerns.   1. The authors need to describe the two datasets in detail. The authors expect us to have blind faith that these two studies represent valid experimental datasets for IPAH. This is the basis for the entire experiment. Why is there such poor overlap between the two datasets (GSE113439 and GSE130391)? Why was NCAPG not identified in the previous assessments of IPAH in the patients described in those datasets? Furthermore, there are other valid datasets that the authors must have been aware of (for example; GSE117261- Xu et al.PMID:33478117). Why wasn't this one also chosen and why did this paper not detect NCAPG as a potential target gene? 2. A major paper on IPAH was published recently (Kariotis et al. Nature Communications, 2021). The authors have failed to acknowledge this paper and other important papers in the field. With regards to the Kariotis paper, the authors of that paper were able to identify three major subgroups of IPAH that accounted for 92% of IPAH patients. This paper is surely worth discussing in the context of the work here. 3. Considering that there are a number of papers that analyze and reanalyze datasets to identify novel targets for pathogenesis or treatment, what is unique about this paper, except that the authors came up with a different target? 4. All 4 genes identified are involved in cell cycle regulation and cell growth. Should we be surprised that knocking down the NCAPG gene results in decreased cell proliferation? This is a no-brainer. Is there perhaps a better experiment to show that NCAPG is actually involved in IPAH, other than it is more highly expressed in IPAH? This is a tenuous link at best.

Round 2

Reviewer 1 Report

Thanks for addressing the comments and explanation. 

Reviewer 2 Report

The authors have addressed all my concerns.